# Q-Herilearn: Assessing heritage learning in digital environments. A mixed approach with factor and IRT models

Olaia Fontal [ID][1]☯*, Alex Ibañez-Etxeberria[2]☯, Víctor B. Arias[3]☯, Benito Arias[4]☯

1 Department of Didactics of musical, plastic and corporal expression, University of Valladolid, Campus Miguel Delibes, Valladolid, Spain, 2 Department of Didactics of mathematics, experimental and social sciences, University of the Basque Country UPV-EHU, San Sebastián, Spain, 3 Department of Personality, assessment and psychological treatments, University of Salamanca, Salamanca, Spain, 4 Department of Psychology, University of Valladolid, Campus Miguel Delibes, Valladolid, Spain

☯ These authors contributed equally to this work.
* olaia.fontal@uva.es

**Data Availability Statement:** All relevant data files, including responses from 1454 participants to the Q-Herilearn scale (link: https://doi.org/10.6084/m9. figshare.25208969.v1) and a list of variables and

## Abstract

The assessment of heritage learning in digital environments lacks instruments that measure it with sufficient guarantees of accuracy, validity, and reliability. This study attempts to fill this gap by developing an instrument that has shown solid metric qualities. The process of design and calibration of a scale applied to 1,454 participants between 19 and 63 years of age is presented in this article. Exploratory factor analysis (Exploratory Structural Equation Modeling ESEM) and Item Response Theory models (Graded Response Model GRM) were used. Sufficient evidence of both reliability and validity based on content and internal structure was obtained. Invariance of scores as a function of gender and age of participants has also been demonstrated. The discrimination parameters of the items have been found to be high, and the test information curves have shown that the subscales measure with sufficient precision wide ranges of the respective latent variables. The instrument presents wide possibilities of application to various areas of Heritage Education (e.g., design of programs in HE, definition and planning of teaching objectives, evaluation of programs, etc., in virtual environments).

## Introduction

In the last decade, digital environments have positioned themselves as burgeoning educational settings for teaching cultural heritage, not only due to their massive use, but also because of the potential they represent for learning in the sphere of heritage education [1]. Digital media are frequently presented as extensions or complements of real physical environments; for this reason, heritage learning outcomes obtained in digital environments are measured in research work in close connection with the geographical context [e.g., 2–5]. Some studies, however, go beyond such spatial references and instead focus on digital environments as specific (informal) heritage learning settings, so that they are understood as stand-alone informal learning environments [6, 7].

values (link: https://doi.org/10.6084/m9.figshare.25212038.v1), are accessible via the figshare repository.

**Funding:** This research has been funded by the Ministry of Science and Innovation, State Research Agency, within the project PID 2019-106539RB-I00, "Learning models in digital environments for heritage education". Principal Investigators: Olaia Fontal Merillas and Alex Ibáñez Etxeberria. This research has been funded by the Ministry of Science and Innovation, Next Generation EU (Recovery, Transformation and Resilience Funds), within the project PDC2022-133460-I00, "Heritage education in Spain in the face of the 2030 agenda: heritage literacy plan in digital environments". Principal Investigators: Olaia Fontal Merillas and Alex Ibáñez Etxeberria. The funders had no role in study design, data collection and analysis, decision to publish, or preparation of the manuscript.

**Competing interests:** The authors have declared that no competing interests exist.

The evaluation of learning in heritage education has been dispersed in terms of the targets of measurement, which cover the upgrading of acquired knowledge [8], the development of competencies [9], sensory-motor learning [10], the learning experience, the enjoyment derived from the latter [11], the attitudes towards heritage [12], and even social learning outcomes [13]. When they rely on previous designs and interventions, studies usually measure the specific effects derived from their implementation [14, 15]. In the particular case of evaluation of technology-mediated learning, the studies deal with the impact of mobile-learning heritage knowledge [16, 17] or the effectiveness of certain technological resources for achieving heritage learning goals [18, 19], including analyzing the quality of learning from a psychoneurological perspective [20].

Among the studies specifically dedicated to the evaluation of heritage learning in digital environments, some are concerned with gauging the effects of intrinsic motivation and competence obtained by means of virtual-reality-based learning, which are compared with traditional text-based learning [21]; it has also become possible to evaluate the potential of portals and other synchronous learning platforms to promote empathy among diverse cultural populations, considering that standard heritage spaces (for example, museums) should adopt synchronous learning to develop a more participatory and dynamic educational model [22]. Along this line, which seeks to combine face-to-face and virtual experience, the cognitive, emotional and social dimensions involved in the learning process have likewise become the object of analysis [23], and so have the processes linked to the transmission of heritage values on social media [24].

Despite the large number of studies related to heritage learning in digital environments, almost all of them put the focus on the implementation of innovation, and have an exploratory nature, with the limitations that this entails in terms of generalizing results. With the exception of the *Instructional Materials Motivation Survey Questionnaire* [25]–which evaluates attention, confidence and satisfaction factors–and the intuitive evaluation system designed by Lee et al., (2016) [26]–which attempts to measure the affective, cognitive and operational dimensions in learning processes–there are no specific studies on any instrument that measure learning outcomes in digital heritage education environments. In the studies collected, ad hoc questionnaires have mostly been used for the specific designs under scrutiny [8, 18, 25] in which no description is provided of the processes of calibration or validation of the scores that were followed. This ad hoc approach makes it difficult to perform reliable comparisons between results from various studies that measure the same concept.

Likewise, the evaluation of learning in digital heritage education environments has not been constructed on the basis of an organized sequence that identifies the main dimensions or latent variables. All of this makes it necessary to deploy a standardized scale, capable of accurately measuring the constructs of a sequence of heritage processes in different contexts, environments and actions. Furthermore, a scale is required that allows results to be compared across different groups and populations, using standard scores to evaluate the effectiveness of different heritage education programs or, where appropriate, measure changes in heritage learning outcomes.

Following from the design and calibration method of the Q-Edutage scale focused on the evaluation of heritage education programs [27], we propose to lay out and calibrate a scale articulated around seven factors underpinned by the seven verbs in the Heritage Learning Sequence (HLS) which define the main learning actions concerning heritage (i.e., knowing, understanding, respecting, valuing, caring, enjoying and transmitting: Fontal et al., 2022 [28]) and make up the seven dimensions of the Q-Herilearn scale that we present here. These terms comprise the educational action that results in heritage learning outcomes in digital environments and are identified following the theoretical model that supports the HLS, in turn

inspired by the content analysis of the main international texts, treaties and recommendations (UN, UNESCO, EU) in matters of heritage [29] as well by the analysis of the main verbs used in the conceptualization of heritage by users of digital environments [28]:

1. Knowing: Acquiring an understanding of the range of cultural assets that are part of the historical and cultural heritage of a society or community.

2. Understanding: Comprehending the meaning of heritage, its historical, cultural and social context, as well as the relationships and connections between different heritage items.

3. Respecting: Adopting an attitude of care, appreciation, commitment and responsibility towards heritage.

4. Valuing: Appreciating the importance and significance of heritage, recognizing its valuable qualities for a community.

5. Caring: Taking action to protect, conserve and preserve heritage for present and future generations.

6. Enjoying: Actively experiencing and appreciating heritage for pleasure and personal enrichment.

7. Transmitting: Effectively sharing and communicating the knowledge, values, traditions, stories and significance of heritage to present and future generations.

### Study goals

As a result of the above considerations, the present study aims to (a) develop an instrument with sound metric qualities that assesses how we learn heritage in digital environments and (b) calibrate the instrument itself by using a mixed approach based on measurement models (Exploratory Etructural Equations Models) and Item Response Theory.

### Research design and hypotheses

This work follows the methodology of cross-sectional survey designs, the essential purpose of which is to provide a quantitative description of participants' opinions as expressed through responses to structured questionnaires [30, 31]. The exploratory study starts from the HLS, which identifies the seven main verbs in heritage learning set out above. These verbs constitute the seven dimensions of the Heritage Process Model (HPM, Fontal et al., 2022 [28]). Each of the latent variables is assessed by means of 7 indicators. Both unidimensional models and an ESEM model consisting of the 49 items and the seven factors or dimensions have been analysed (see Fig 1). The hypotheses are derived directly from these models, and are as follows: (a) each of the dimensions (knowing, understanding, respecting, valuing, caring, enjoying, transmitting) is measured by 7 indicators, as depicted in Fig 1A, and (b) the indicator loadings will be significant and higher on each reference factor than on the rest of the factors, as shown in Fig 1B.

## Materials and methods

### Participants

The final sample consisted of $N = 1,454$ participants aged 19 to 63 years ($M = 26.71$, $SD = 10.51$). For some of the analyses below, the variable age was categorized into six groups. The defining characteristics of the participants (age, gender, country of residence, number of countries visited, area of residence, mother tongue, level of education) are summarized in Table 1.

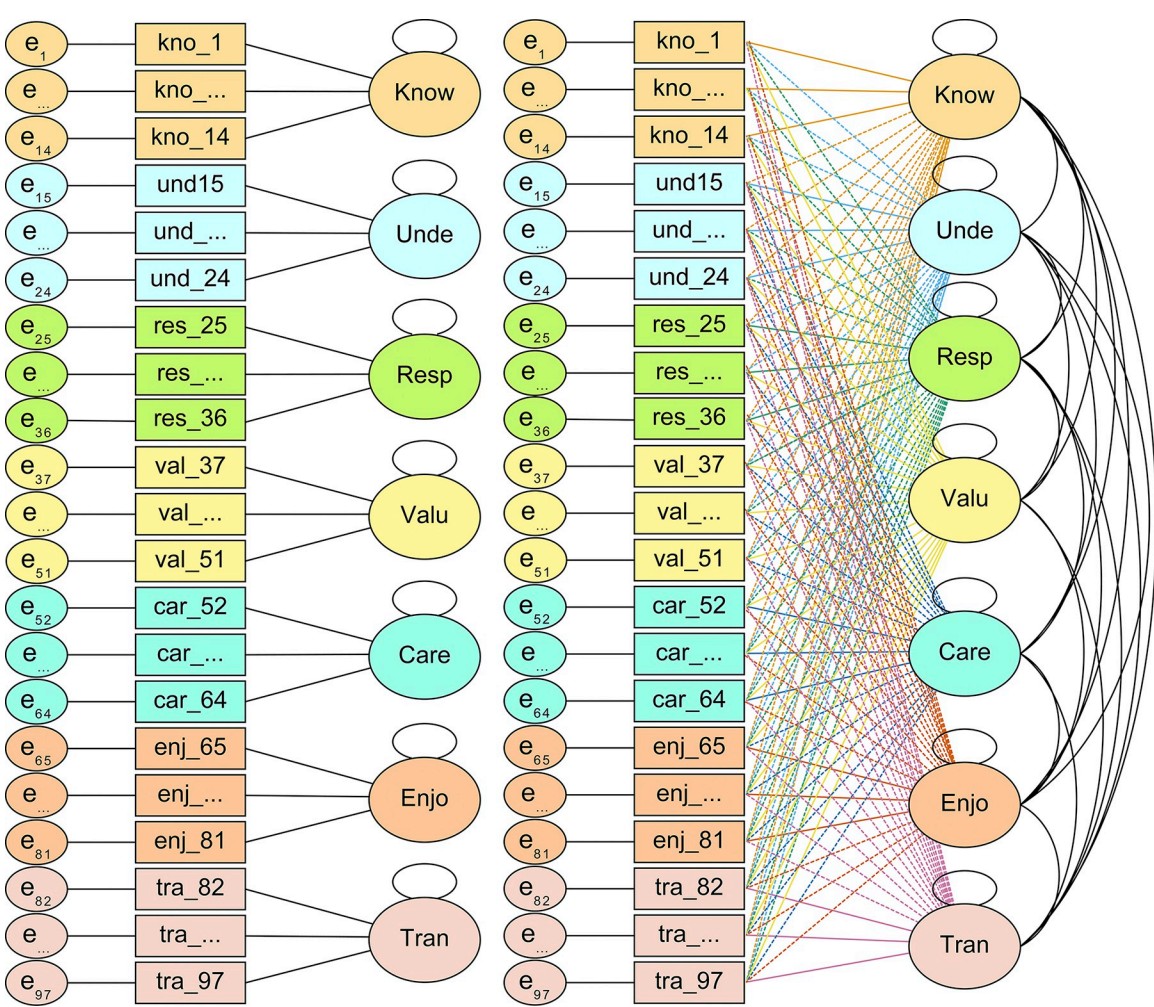

**Fig 1. Schematic representation of the one-dimensional and ESEM models.** A Unidimensional models. B ESEM model.

Participants were predominantly under the age of 30 (85.6%), female (69.8%) and residents in Spain (90.1%), living in urban areas (79.4%), with Spanish as their mother tongue (80.3%) and with a higher education background (85.6%).

All participants completed an online survey (https://oepe.es/escala-herilearn/) between May 9, 2022 and September 2, 2023, after being informed of the purposes of the study and guaranteed complete data confidentiality, in accordance with the provisions of the CEISH UPV-EHU Ethics Committee (Cod: M10_2021_31). They were also informed that the survey consisted of 97 items. Participants could interrupt, postpone or abandon the survey at any time (in the latter case, the data were automatically deleted). A total of 1,389 responses were obtained with complete socio-demographic information, plus 65 in which only some of the fields were filled out. Acceptance of informed consent was a prerequisite for responding to the survey.

## Sample size, power and precision

In order to determine the minimum sample size, we took into account (a) statistical power (at least 80%); (b) effect size ($f^2 \geq .35$) and (c) significance level ($\alpha = .05$). To calibrate the

**Table 1. Sample characteristics.**

| Variable | Value | Frequency | % | Valid % |
|---|---|---|---|---|
| **Age** | Less than 20 | 389 | 26.8 | 27.8 |
| (M = 26.71, | 20–30 | 808 | 55.6 | 57.8 |
| SD = 10.51) | 31–40 | 59 | 4.1 | 4.2 |
| | 41–50 | 73 | 5 | 5.2 |
| | 51–60 | 39 | 2.7 | 2.8 |
| | More than 60 | 30 | 2.1 | 2.1 |
| | **Total** | 1,398 | 96.1 | 100 |
| **Gender** | Female | 976 | 67.1 | 69.8 |
| | Male | 410 | 28.2 | 29.3 |
| | Non-Binary | 13 | 0.9 | 0.9 |
| | **Total** | 1,399 | 100 | 100 |
| **Residence** | Spain | 1,254 | 86.2 | 90.1 |
| | Mexico | 107 | 7.4 | 7.7 |
| | Other | 31 | 2.1 | 2.2 |
| | **Total** | 1,392 | 95.7 | 100 |
| **Countries** | None | 145 | 10 | 10.4 |
| **visited** | 1 to 3 | 645 | 44.4 | 46.4 |
| | 4 to 7 | 402 | 27.6 | 28.9 |
| | 8 or more | 197 | 13.5 | 14.2 |
| | **Total** | 1,389 | 95.5 | 100 |
| **Area** | Rural area | 288 | 19.8 | 20.6 |
| | City or urban area | 1,108 | 76.2 | 79.4 |
| | **Total** | 1,396 | 96 | 100 |
| **Language** | Spanish | 1,121 | 77.1 | 80.3 |
| | Basque | 168 | 11.6 | 12 |
| | Double | 45 | 3.1 | 3.2 |
| | Other | 62 | 4.3 | 4.4 |
| | **Total** | 1,396 | 96 | 100 |
| **Studies** | Primary Education | 6 | 0.4 | 0.4 |
| | Secondary Education | 129 | 8.9 | 9.2 |
| | Vocational Education | 66 | 4.5 | 4.7 |
| | University Education | 1,195 | 82.2 | 85.6 |
| | **Total** | 1,396 | 96 | 100 |

precision and power achieved by the analysis given the sample size used ($N = 1,328$), we performed a Monte Carlo analysis (10,000 replicates) using as population parameters the results of the structural model (see Supporting Information, S9 Table), as recommended by Muthén & Muthén (2002) [32].

The analysis was performed with Mplus, v. 8.10 [33], and convergence was achieved without problems in 100% of the requested replicates. S9 Table (Supporting Information) shows the results on the parameters of the structural model. The population parameters and the means of the parameters estimated by the model were very similar in all cases, suggesting the absence of bias in the estimation. Similar results were observed in the estimation of the standard error, with no evidence of relevant bias in any of the parameters analyzed. The Mean Squared Error values (MSE) were in all cases very close to zero, confirming the absence of bias observed in the comparison between population and simulated parameters. Between 94% and 96% of the replicates contained a population value with a 95% confidence interval. For

population parameters greater than zero, the test reached the maximum power (1,000) in all cases. For population parameters with a value of zero, the proportion of replicates in which the parameter was significant always remained close to the desired value of .05. In conclusion, the results of the Monte Carlo analysis suggest that with this sample size very precise estimates of the model parameters were achieved, with high power and a low probability of Type I error.

## Procedure

### Data collection and cleaning

Data were retrieved from the LimeSurvey platform, transferred to R and cleaned using the following three strategies: outlier filtering, multivariate outlier detection and missing data processing.

### Selection of anomalous responses

Anomalous response patterns (e.g., repetitive, invariant, random or sloppy responses) can profoundly alter the results of data analysis, even if they occur in very small proportions [34, 35]. To avoid this bias, the data were cleaned in two ways: first, we eliminated cases where the same answer was given to all 49 items (Straight Lining) ($N = 9 = 0.62\%$), considering that this pattern is highly improbable given the number of test items. Secondly, we estimated the polytomous mode of the standardized likelihood ratio $l^P{}_z$ [36, 37] for each response vector. Extreme values in the left tail of the $l^P{}_z$ ($\leq$ -3) indicate highly unexpected response patterns, as predicted by the measurement model: these patterns are usually the result of random responses not based on item content. The cut-off point set at -1.6308 identified 71 cases (4.88%) with anomalous responses. The more conservative cut-off point of -3.00 identified 26 anomalous responses (1.79%), as seen in Fig 2, which shows the histogram and density of PFS (Person Fit Scores). Consequently, the 26 cases with $l^P{}_z \leq$ -3 were excluded from further analysis.

### Detection of multivariate outliers

As shown in Fig 3, we plot the robust Mahalanobis squared ordered distances of the observations against the empirical distribution function of $MD^2{}_i$. Fig 3A shows the maximum value curve of the $MD^2{}_i$, distribution, while Fig 3B shows the maximum values detected by the specified quantile (97.5%). Multivariate outliers, i.e. observations outside the 97.5 quantile of the $\chi^2$ distribution ($N = 26$, 1.79%) marked in red in Fig 3B (the numbers correspond to the observations in the original database) were removed. The first subfigure shows the peak value curve of the $MD^2{}_i$ distribution, and the second subfigure shows the peak values detected by the specified fitted quantile (97.5%).

In summary, the combination of the procedures described above resulted in 8.67% of participants ($N = 126$) meeting one of the selection criteria and therefore being removed from the database for further analysis. S10 Table (Supporting Information) provides a summary of the eliminated cases.

### Treatment of missing data

Given the sufficient sample size, the low proportion of cases with missing data ($< 3\%$), the high average data coverage ($> 98\%$) and the MCAR structure (Little's test: $\chi^2{}_{(5056)} = 5212.605$, $p = .061$), multiple imputation was considered unnecessary and Full Information Maximum Likelihood (FIML) was used to estimate the parameters of the factor models, using all available data [38].

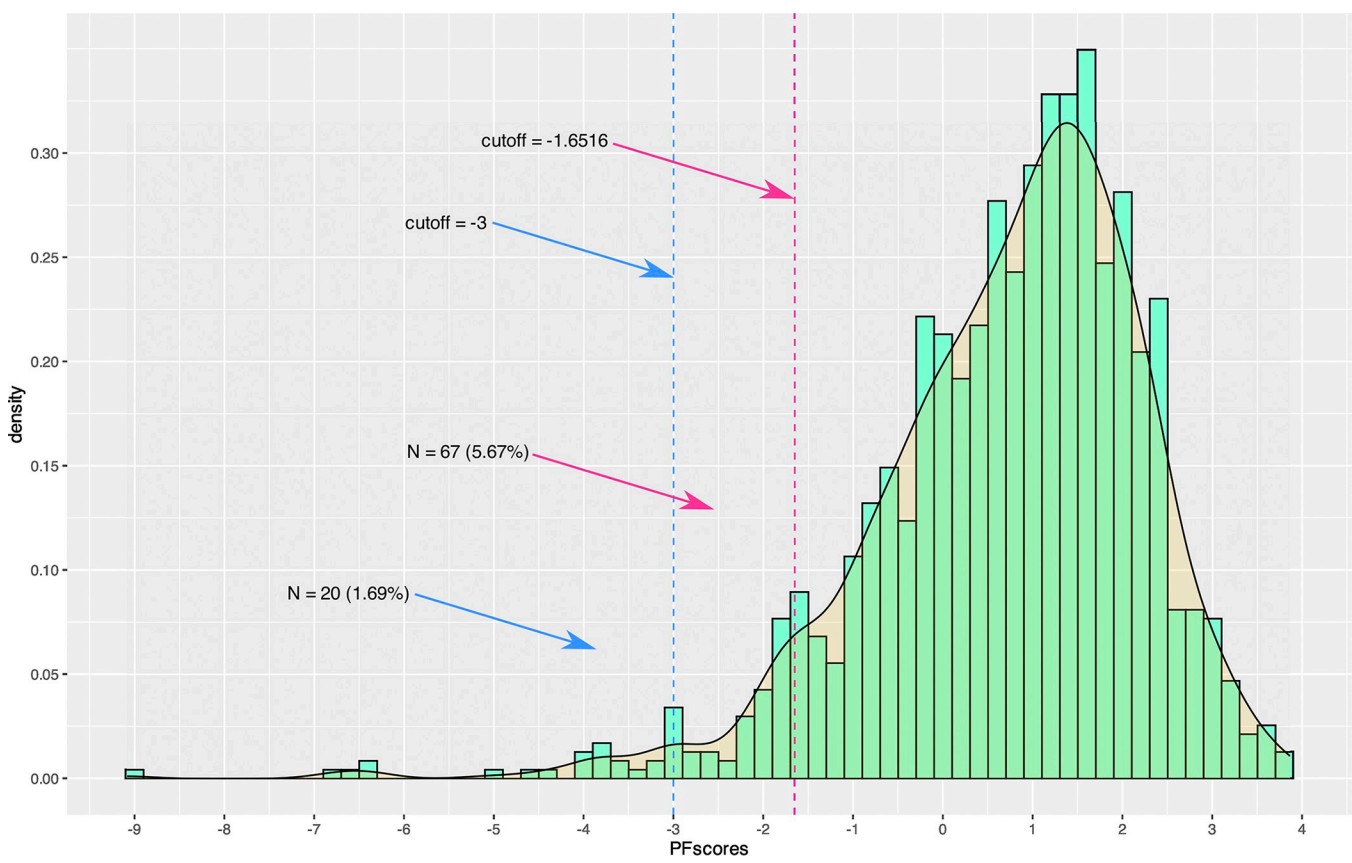

**Fig 2. PFScores.** Cutoff = -1.73 and -3.00.

### Data analysis

Analysis procedures

Two types of analysis have been employed: (a) factor analysis and (b) analysis using Item Response Theory models.

*Factor analysis.* Factor analysis was conducted along three phases. The aim of the first phase was to estimate the fit of each subscale to the one-dimensional confirmatory factor model, so

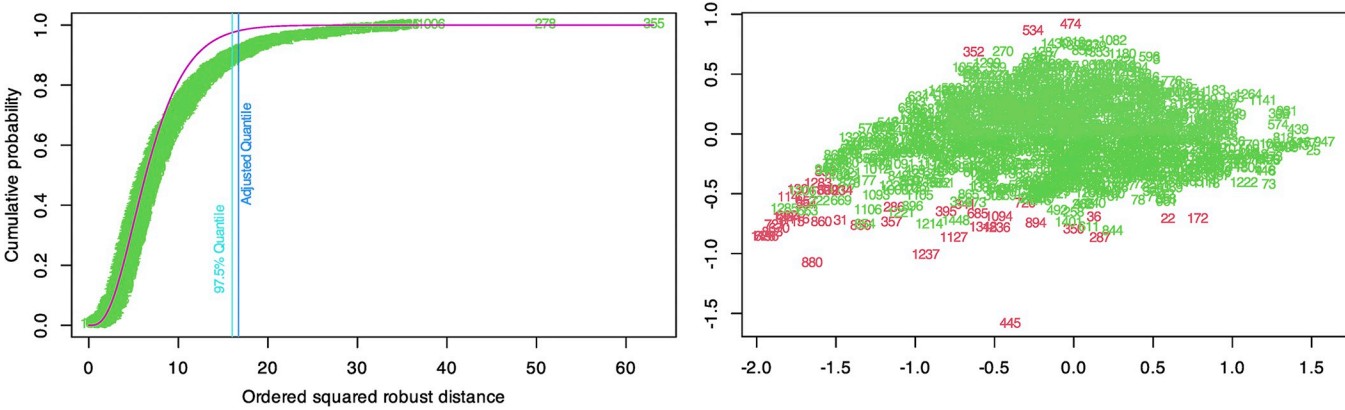

**Fig 3. Multivariate outliers detection.** A Outliers (97.5% quantile and adjusted quantile). B Outliers based on 97.5% quantile.

as to estimate the convergent validity of the items, and to verify that each subscale acquired sufficient reliability and internal consistency. To this purpose, seven unidimensional confirmatory models were estimated for each factor (see Fig 1A), as well as the average variance extracted (AVE), Cronbach's ordinal alpha, McDonald's omega, composite reliability (CR) and the Great Lower Bound of Reliability (GLB).

The aim of the second phase of the analysis was to investigate whether (a) it is possible to recover the theoretical structure of the measure from the pooled data, and (b) the items have sufficient discriminative ability, i.e., they measure their theoretical factor substantially better than the rest of the factors. For this purpose, an exploratory structural equation model (ESEM; [39, 40]) was estimated using all items of the scale simultaneously (Fig 1B). Oblique target rotation was used. Target rotation allows items to load freely on their reference factor, and seeks the rotated solution where the cross-loadings are as close as possible to the expected size according to the theoretical starting model (in this case, zero). Thus, by allowing the expression of a priori hypotheses about the pattern of primary loadings and cross-loadings, the target rotation allows the ESEM to be used in a semi-confirmatory way [39].

The aim of the third phase was to verify compliance with measurement invariance by gender and age. In the case of gender, two nested ESEM models were estimated [41]: configural (equivalence of number and layout of factors), and scalar (equivalence of primary loadings and cross-loadings, and of thresholds). In the case of age, it being a continuous variable, we chose an approach based on multiple indicator multiple cause models (MIMIC; [42]), following the recommendations of Morin et al. (2016) [43], to assess invariance by comparing the fit of nested MIMIC models. With age as the predictor variable, two models were compared: (a) an invariant model, where regression coefficients between age and each of the factors were estimated, restricting any direct correlation between age and item responses to zero, and (b) a saturated model, which assumes no scalar invariance, restricting any correlation between age and the factors, and estimating regression coefficients between age and each of the items. If the fit of the invariant (more parsimonious) model is similar to the fit of the saturated model, one can with reasonable confidence rule out the presence of serious violations of scalar invariance.

All factor models were estimated using Weighted Least Squares Mean and Variance Adjusted (WLSMV), given the ordinal nature of the item responses [44]. Goodness of fit was assessed using the Comparative Fit Index (CFI), the Tucker-Lewis Index (TLI), and the Root Mean Square Error of Approximation (RMSEA). Conventionally, CFI and TLI values above .90 and .95 respectively indicate acceptable and good fit, [45, 46]. In the case of RMSEA, values at or below .05 and .08 are respectively considered good and acceptable [47].

In order to make decisions on the significance of differences in fit between nested models, we followed the recommendations of Chen (2007) [48] and Cheung & Rensvold (2002) [49], according to which increases of less than .01 in CFI and TLI, and decreases of less than .015 in RMSEA suggest that there is no relevant change in the fit of one model with respect to the next most restrictive one. In addition, maximum likelihood with robust standard errors (MLR) was applied on the data treated as categorical variables to estimate the Bayesian Information Criterion (BIC) and the Akaike Information Criterion (AIC): when comparing two nested models, lower values of BIC and AIC suggest a better fit.

*Analysis using item response theory models.* Once the structure of the data had been analysed, we conducted a detailed analysis of the items' properties by estimating IRT models. As a preliminary step, we investigated the dimensionality of each theoretical factor in order to ensure that the data were suitable for analysis using unidimensional IRT models. In order to secure sufficient compliance with unidimensionality and conditional independence, each scale had to meet the following requirements: (a) the percentage of variance explained by the second factor should not exceed that explained by random data simulated by optimized parallel

analysis [50]; (b) the Explained Common Variance ECV of the first factor should be greater than. 80; (c) the Mean of Items Residuals Absolute Loadings (MIREAL; [51]) should be less than .3; (c) the Factor Determinacy Index [51] (FDI; [51]) should be greater than .90; and (d) the Generalized H Index (G-H; [52]) should be greater than .80.

ECV measures the dominance of the first factor over the rest of the factors. Values above .80 allow us to conclude that the solution is essentially unidimensional [53]. MIREAL is the mean of the absolute loadings on the second factor MRFA (Minimum Rank Factor Analysis), and assesses the extent to which the structure of the data deviates from unidimensionality. As a practical rule, values above .30 indicate the absence of a relevant residual factor. FDI is the correlation between factor score estimates and the levels of the latent factors they estimate [54]. Values above .80 are acceptable. Finally, G-H measures the degree to which a factor is correctly represented by a set of items, i.e., the maximum proportion of factor variance that can be explained by its indicators (construct reliability), with values above .70 being acceptable.

To calculate the indices described above, we estimated an exploratory bifactor model for each facet using Minimum Rank Factor Analysis.

After ensuring that all factors reached a sufficient degree of unidimensionality and conditional independence, we estimated a Graded Response Model (GRM; [55]) for each dimension. We then inspected the discrimination and difficulty parameters for each item, as well as the information functions of the test.

Dimensionality analyses were performed with the FACTOR, v. 12.04.04 software [56]. IRT analyses were performed using Mplus, v. 8.10 [33].

## Instrumentation

**Item development and first review.** The Q-Herilearn scale is a probability scale of summative estimates that measures different aspects of the learning process in Heritage Education. It consists of the seven factors (Knowing, Understanding, Respecting, Valuing, Caring, Enjoying and Transmitting) defined in the introduction to this paper. Each dimension is measured by seven indicators scored on a 4-point frequency response scale (*1 = Never or almost never; 2 = Sometimes; 3 = Quite often; 4 = Always or almost always*).

In order to examine the literature on the topic published in recent years, a WoS search was carried out (March 2022). 212 references were found using the following search terms: "heritage AND (evaluat* OR assessment OR scal*) in Title. Document Types: Article. Database: Web of Science Core Collection. Publication Years: 2010 to 2023. Research Areas: Arts Humanities Other Topics or Social Sciences Other Topics or Psychology." As mentioned in the introduction, none of the works retrieved were dedicated to developing specific instruments for assessing heritage learning in digital contexts.

Therefore, in view of the lack of instruments, and putting the focus on the concepts included in the heritage sequence [57], a pool of items was drawn up to measure each of the seven dimensions of the sequence.

In the design and general implementation of the instrument, we followed the common postulates and recommendations for the development of scales and assessment instruments. In the wording of the items, we followed the usual rules in the construction of items of probabilistic scales for summative estimates [58–60]: (a) item content should refer to the present; (b) item content should not refer to facts unrelated to the respondent; (c) item content should have only one interpretation; (d) item content should be relevant to the dimension it is intended to measure; (f) avoid extreme statements (i.e., statements that can be endorsed by almost everyone or almost no one); (g) items should cover the full range of each dimension;

(h) items should be written in clear, simple, straightforward language; (i) sentences should be short (i.e., they should not exceed 20 words); (j) each sentence should contain only one complete idea; (k) statements containing extreme expressions such as "all", "always", "none" or "never" should be avoided; (l) items should not contain adverbs such as "only", "solely", "merely" or similar ones; (m) statements should be formulated in simple rather than compound or complex sentences; (n) vocabulary should be accessible to potential respondents; (o) item valence should be positive; and (p) items should not contain negative or double negative expressions.

## Results

### Evidence of validity

**Content-based validity evidence.**   In order to ensure the relationship between the content of the instrument and the construct it was intended to measure [58], both logical (clarification of the content through focus groups) and empirical (submission of the items to expert judgment, as detailed below) analyses were carried out.

Following the recommendations mentioned above, an initial pool of 117 items was drawn up, the content of which was submitted to 40 independent expert judges/raters, who had to evaluate on a scale of 1 to 4 points (a) the clarity of the item formulation; (b) the relevance or importance of each item for measuring the dimensions of the sequence; and (c) the suitability or appropriateness for measuring these dimensions. In addition, the judges had to indicate to which of the seven theoretical dimensions each item could be ascribed on the basis of its content. The judges issued their ratings online, through the LimeSurvey platform, during the second half of May 2022 (the matrix of judges' ratings can be found in the Supporting Information).

During first screening analysis, items with a mean lower than 3 and a standard deviation higher than 1 according to the rating given by the judges were discarded. This first screening resulted in a set of 97 items that met these requirements (see Supporting Information, S1–S7 Tables).

Inter-rater agreement was then calculated using four procedures: (a) Fleiss' kappa [61], (b) observed global agreement [62], (c) Krippendorff's alpha [63, 64], and (d) Bangdiwala coefficients [65] (Bangdiwala & Shankar, 2013).

The results of the analysis of the agreement matrices using the $B_N$ coefficients for nominal data and the $B^w_N$ coefficients for ordinal data from Bangdiwala (Bangdiwala & Shankar, 2013) are shown in Fig 4. The overall coefficients of agreement can be considered very satisfactory. Thus, the degree of agreement was almost perfect in relevance ($B^w_N$ = .811) and appropriateness ($B^w_N$ = .808), and substantial in clarity ($B^w_N$ = .772) and dimension ($B_N$ = .539), in accordance with the interpretation guidelines proposed by Muñoz and Bangdiwala (1997, p. 111) [66].

The agreement of the 3,880 decisions made by the judges in their ascription of the items to each of the seven dimensions resulted in a Fleiss Kappa value of κ = .671; the observed agreement, OA = .722 and the Krippendorff alpha coefficient, α = .671 (see complete results in Supporting Information, S11–S17 Tables). Taking into account the magnitude of the aforementioned coefficients, the overall agreement among the judges can be considered substantial [66, 67].

**Evidence based on internal structure.**   *Factor structure.* Table 2 shows the results of the factor analysis (the polychoric correlations among items are delineated in S8 Table of the Supporting Information). Each unidimensional baseline model was estimated with 14 degrees of freedom; the two additional free parameters of each model reported in Table 2 correspond to

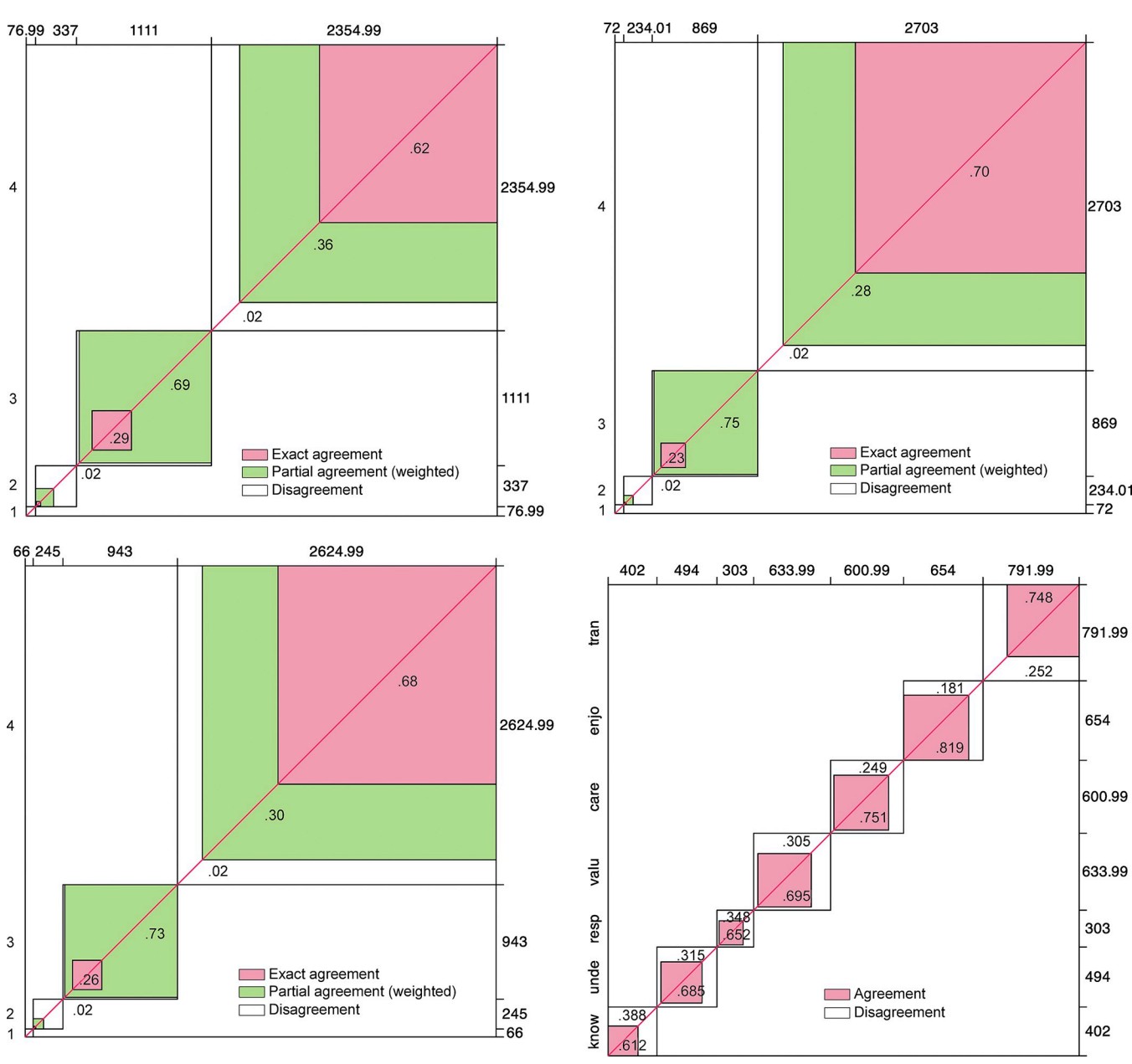

**Fig 4. Results of the inter-rater agreement analysis.** A Clarity ($B^w_N$ = .772). B Relevance ($B^w_N$ = .811). C Adequacy ($B^w_N$ = .808). D Dimension ($B_N$ = .539).

the estimation of two correlations between the residuals of pairs of items that, showing clear semantic similarity, obtained MI (Modification Index) and SEPC (Standardized Expected Parameter Change) substantially greater than 10 and 0.3, respectively. The fit of the unidimensional models was reasonably high, with RMSEA values between .086 (RES scale) and .037 (ENJ scale), CFI values between .983 (RES scale) and .998 (ENJ scale), and SRMR between .029 (RES scale) and .011 (ENJ scale).

*Convergent and discriminant evidence.* Table 3 shows the reliability and internal consistency estimators from raw scores (Cronbach's alpha) and unidimensional models (McDonald's omega and GLB), as well as the composite reliability (CR) and the item convergent validity estimator (AVE). All alpha, omega, GLB and CR values were above .80, with the minimum

**Table 2. Model's fit.**

| Model | FP | RMSEA (CI) | CFI | TLI | $\chi^2$ | DF | SRMR |
|---|---|---|---|---|---|---|---|
| KNO | 30 | .065 (.052; .079) | .993 | .989 | 82 | 12 | .020 |
| UND | 30 | .053 (.040; .067) | .994 | .990 | 57 | 12 | .016 |
| RES | 30 | .086 (.073; .100) | .983 | .968 | 123 | 12 | .029 |
| VAL | 30 | .072 (.059; .086) | .990 | .983 | 98 | 12 | .020 |
| CAR | 30 | .049 (.035; .063) | .997 | .995 | 50 | 12 | .013 |
| ENJ | 30 | .037 (.023; .052) | .998 | .997 | 34 | 12 | .011 |
| TRA | 30 | .073 (.060; .087) | .992 | .986 | 100 | 12 | .020 |
| ESEM 7 factors | 469 | .036 (.034; .037) | .977 | .968 | 2357 | 1176 | .020 |

*Note*. KNO = Knowing; UND = Understanding; RES = Respecting; VAL = Valuing; CAR = Caring; ENJ = Enjoying; TRA = Transmitting

being observed for the RES scale ($\alpha$ = .83, $\omega$ = .83, *GLB* = .87, *CR* = .87) and the maximum for the ENJ scale ($\alpha$ = .90, $\omega$ = .89, *GLB* = .91, *CR* = .92). The AVE values were satisfactory in all cases except for the RES factor, with an *AVE* value = .48, very close to the minimum value necessary (.50) to guarantee the convergent validity of the factor. It should be noted, in any case, that the value .50 is within the limits of the confidence interval used.

The ESEM model showed a reasonably high fit (*RMSEA* = .036; *CFI* = .977; *SRMR* = .020). However, this result was to be expected given the high parameterization of the model. Table 4 shows the standardized factor loadings, and the Item Explained Common Variance (iECV). The iECV quantifies the variance captured by the item in its reference factor, versus the amount of common variance captured by all possible cross-loadings. Accordingly, here we use the iECV as an estimator of the item's ability to discriminate between its theoretical membership factor and all other factors, with a minimum desirable value of .50 (an *iECV* $\geq$ .50 indicates that the primary factor explains as much or more common variance in item responses than all other factors combined).

Regarding the value of the primary loadings and cross-loadings, it is observed in the first placed that the model has satisfactorily recovered the theoretical structure, given that in all items the most salient loading is always the one corresponding to the primary factor (see Fig 5). Secondly, the iECV values were in a range between .374 (*res26*) and .977 (*res30*), with 45 of the 49 items showing a value above .50. In conclusion, it was possible to reproduce from the data a structure highly consistent with that expected by the theoretical model, without the need to eliminate items or introduce modifications into the model specification.

The correlations between the factors (S18 Table, Supporting Information) were adequate in all cases, ranging from -.075 (RES-CAR) to .602 (VAL-UND).

*Invariance analysis*. Tables 5 and 6 show the results of the invariance analysis by gender and age.

**Invariance by gender.** Regarding gender, differences in favor of the scalar model were observed in all the indices (*RMSEA* = -.009; $\Delta CFI$ = .007; $\Delta TLI$ = .013; $\Delta AIC$ = -216; $\Delta BIC$ = -1973), except in SRMR, with a slight difference in favor of the configural model ($\Delta SRMR$ = -.004). This result suggests the absence of substantial differences in the model parameters according to the gender of participants. The category "non-binary" has not been included in this analysis due to the low number of participants (*N* = 13) who indicated this option.

**Invariance by age.** With respect to age, the saturated model obtained a slightly better fit ($\Delta RMSEA$ = .001; $\Delta CFI$ = -.001; $\Delta AIC$ = 75; $\Delta BIC$ = 1345; $\Delta SRMR$ = .004). We further investigated the local fit of the invariant model in order to detect regression parameters between age and each item that, when set to zero, would reveal a relevant misspecification. However, we

**Table 3. Reliability analysis.**

| Factor | Estimate | McDonald's ω | std. Cronbach's α | GLB | AIC | CR | AVE |
|---|---|---|---|---|---|---|---|
| KNO | Point estimate | .88 | .88 | .91 | .51 | .91 | .58 |
| | 95% CI lower bound | .87 | .87 | .90 | .49 | .90 | .56 |
| | 95% CI upper bound | .89 | .89 | .91 | .53 | .91 | .60 |
| UND | Point estimate | .85 | .85 | .88 | .45 | .89 | .50 |
| | 95% CI lower bound | .84 | .84 | .87 | .43 | .88 | .49 |
| | 95% CI upper bound | .86 | .86 | .90 | .48 | .90 | .55 |
| RES | Point estimate | .83 | .83 | .87 | .41 | .87 | .48 |
| | 95% CI lower bound | .81 | .82 | .85 | .38 | .86 | .46 |
| | 95% CI upper bound | .84 | .84 | .88 | .43 | .88 | .51 |
| VAL | Point estimate | .86 | .86 | .89 | .46 | .91 | .52 |
| | 95% CI lower bound | .85 | .85 | .88 | .44 | .90 | .51 |
| | 95% CI upper bound | .87 | .87 | .90 | .49 | .92 | .53 |
| CAR | Point estimate | .89 | .89 | .92 | .54 | .89 | .64 |
| | 95% CI lower bound | .88 | .88 | .91 | .51 | .89 | .62 |
| | 95% CI upper bound | .90 | .90 | .93 | .56 | .90 | .66 |
| ENJ | Point estimate | .90 | .89 | .91 | .55 | .92 | .62 |
| | 95% CI lower bound | .89 | .89 | .91 | .52 | .92 | .60 |
| | 95% CI upper bound | .90 | .90 | .92 | .57 | .93 | .65 |
| TRA | Point estimate | .88 | .88 | .91 | .51 | .91 | .58 |
| | 95% CI lower bound | .87 | .87 | .90 | .48 | .90 | .56 |
| | 95% CI upper bound | .89 | .89 | .92 | .53 | .92 | .60 |

*Note.* GLB = Greatest Lower Bound Reliability; AIC = Average interitem correlation; CR = Composite Reliability; AVE = Average Variance Extracted

found no clear evidence that the misfit of the invariant model was caused by a particular subset of items, but rather by the accumulation of low magnitude misfits spread across all restricted parameters. Given these results, and the small size of the differences in fit between the invariant and the saturated model, we chose to attribute the differences in fit to a greater parameterization of the saturated model, and not to the presence of relevant invariance problems.

**IRT analysis.** Table 7 shows the parameters obtained after estimation of the seven GRM models. The α discrimination parameters ranged from 1.236 (*res26*) to 3.430 (*tra86*). According to the classification proposed by Baker and Kim (2017) [68], one item obtained a discrimination parameter of moderate size (1.236), six items of high size (between 1.457 and 1.675), and 42 items of very high size (between 1.691 and 3.430). The β parameters were generally adequate, covering in all items a sufficiently wide theta range. However, item *res30* ("I have a respectful attitude towards the diversity of personal heritages") showed an extremely low $\beta_1$ value ($\beta_1 = -5.752$), indicating that this item is extremely "easy" given the characteristics of the sample. Other items showed results opposite to the one described, with very high $\beta_1$ values. This effect was mostly concentrated in the CAR scale. For example, item *car60* ("I collaborate in action networks for the protection of heritage and to prevent the dangers of not taking care of it") showed values $\beta_1 = 1.026$, $\beta_2 = 4.193$, and $\beta_3 = 6.603$. This implies that it is very unlikely to observe an affirmative response ("sometimes" or higher), except in people who show a substantially high level of commitment to active heritage care.

Next, we examined the behavior of each scale by inspecting the Test Information Curves (TICs) depicted in panels (a) through (g) of Fig 6. The KNO, UND, VAL, and ENJ scales were maximally informative over a wide range of the latent variable, ranging from approximately -1.5 to 1.5 standard deviations around the mean. This result suggests that the scales measured

**Table 4. ESEM parameters.**

| item/factor | KNO | UND | RES | VAL | CAR | ENJ | TRA | iECV |
|---|---|---|---|---|---|---|---|---|
| KNO1 | **.631** | .061 | .110 | .003 | .018 | .096 | -.059 | .93 |
| KNO4 | **.492** | .336 | .087 | .035 | -.168 | -.067 | .156 | .58 |
| KNO6 | **.621** | .206 | -.001 | .080 | -.123 | .038 | .130 | .82 |
| KNO9 | **.623** | -.025 | -.106 | .118 | .075 | .025 | .136 | .88 |
| KNO10 | **.675** | -.044 | .082 | -.056 | .208 | .157 | -.115 | .83 |
| KNO11 | **.586** | .057 | .157 | .013 | .024 | .133 | -.016 | .88 |
| KNO13 | **.690** | .127 | -.037 | .031 | .029 | .071 | .045 | .95 |
| UND15 | .356 | **.430** | .026 | -.009 | -.041 | -.052 | .101 | .57 |
| UND17 | .235 | **.564** | .031 | .078 | .032 | .077 | -.058 | .81 |
| UND20 | .159 | **.503** | -.082 | .144 | .153 | .068 | -.083 | .74 |
| UND21 | .208 | **.432** | .146 | .036 | -.146 | .009 | .095 | .66 |
| UND22 | .208 | **.441** | .066 | .170 | .051 | -.049 | .083 | .69 |
| UND23 | -.118 | **.770** | -.030 | -.019 | .126 | .023 | .087 | .94 |
| UND24 | -.180 | **.777** | .043 | -.011 | -.019 | .164 | -.018 | .91 |
| RES26 | .137 | .219 | **.280** | .187 | .139 | .025 | -.099 | .37 |
| RES29 | .078 | .089 | **.598** | -.140 | .135 | .065 | .070 | .85 |
| RES30 | .058 | -.036 | **.894** | -.059 | -.048 | .046 | .077 | .98 |
| RES32 | .032 | .057 | **.598** | .211 | .158 | .030 | -.081 | .81 |
| RES33 | -.015 | .059 | **.575** | .212 | -.088 | -.118 | .062 | .82 |
| RES34 | .000 | .032 | **.529** | .261 | .201 | -.051 | -.061 | .71 |
| RES36 | .013 | -.025 | **.647** | .039 | -.173 | .108 | .037 | .90 |
| VAL43 | .095 | .072 | .178 | **.284** | .266 | .053 | .021 | .40 |
| VAL45 | .049 | .129 | .252 | **.380** | .020 | .155 | -.054 | .57 |
| VAL46 | -.034 | .055 | .322 | **.401** | -.001 | .179 | -.045 | .53 |
| VAL48 | .067 | -.044 | -.045 | **.822** | -.037 | -.069 | .125 | .96 |
| VAL49 | .038 | -.063 | -.052 | **.916** | -.115 | .024 | .031 | .97 |
| VAL50 | -.107 | .167 | .064 | **.516** | .066 | .075 | .017 | .83 |
| VAL51 | -.081 | .091 | .008 | **.551** | .040 | .161 | .092 | .86 |
| CAR56 | -.039 | .042 | .146 | -.033 | **.758** | -.040 | .114 | .93 |
| CAR57 | -.091 | .109 | .036 | .037 | **.770** | .005 | .080 | .95 |
| CAR58 | .059 | -.016 | -.150 | .088 | **.802** | -.036 | .039 | .95 |
| CAR59 | .041 | .086 | .007 | -.064 | **.713** | .057 | .134 | .94 |
| CAR60 | .062 | .006 | -.070 | .041 | **.773** | -.019 | .097 | .97 |
| CAR63 | .078 | -.009 | .129 | .028 | **.521** | .140 | .200 | .77 |
| CAR64 | .058 | .008 | .140 | .080 | **.498** | .190 | .102 | .77 |
| ENJ67 | .177 | -.001 | .081 | .043 | -.013 | **.588** | .062 | .89 |
| ENJ71 | -.085 | .170 | -.051 | .085 | -.042 | **.772** | .029 | .92 |
| ENJ74 | -.111 | .236 | -.068 | .032 | .026 | **.787** | -.007 | .89 |
| ENJ76 | .298 | -.098 | -.194 | .045 | .235 | **.511** | .057 | .57 |
| ENJ77 | .075 | .032 | -.005 | .051 | .035 | **.631** | .003 | .97 |
| ENJ80 | .053 | -.096 | .133 | .057 | -.109 | **.654** | .185 | .84 |
| ENJ81 | .017 | -.057 | .122 | .104 | -.033 | **.714** | .079 | .93 |
| TRA84 | .001 | .038 | .152 | -.030 | -.025 | .187 | **.657** | .88 |
| TRA86 | -.063 | .014 | .056 | -.032 | .117 | .036 | **.819** | .97 |
| TRA87 | -.044 | .063 | -.019 | .049 | -.028 | -.118 | **.885** | .97 |
| TRA89 | .125 | -.052 | -.001 | .017 | .104 | .207 | **.508** | .78 |
| TRA90 | .178 | -.039 | -.193 | .051 | .249 | .240 | **.371** | .42 |

*(Continued)*

**Table 4.** (Continued)

| item/factor | KNO | UND | RES | VAL | CAR | ENJ | TRA | iECV |
|---|---|---|---|---|---|---|---|---|
| TRA96 | -.030 | .034 | .051 | .070 | -.063 | -.077 | **.825** | .97 |
| TRA97 | .044 | -.048 | -.197 | .126 | .375 | .031 | **.429** | .48 |

*Note.* In bold = primary loadings (targeted); iECV = item explained common variance

their respective constructs quite reliably in people with low, medium, and high levels of the latent variable. The TRA scale showed a slightly right-shifted TIC, with maximum information in a range between approximately -0.5 and 1.5 theta values.

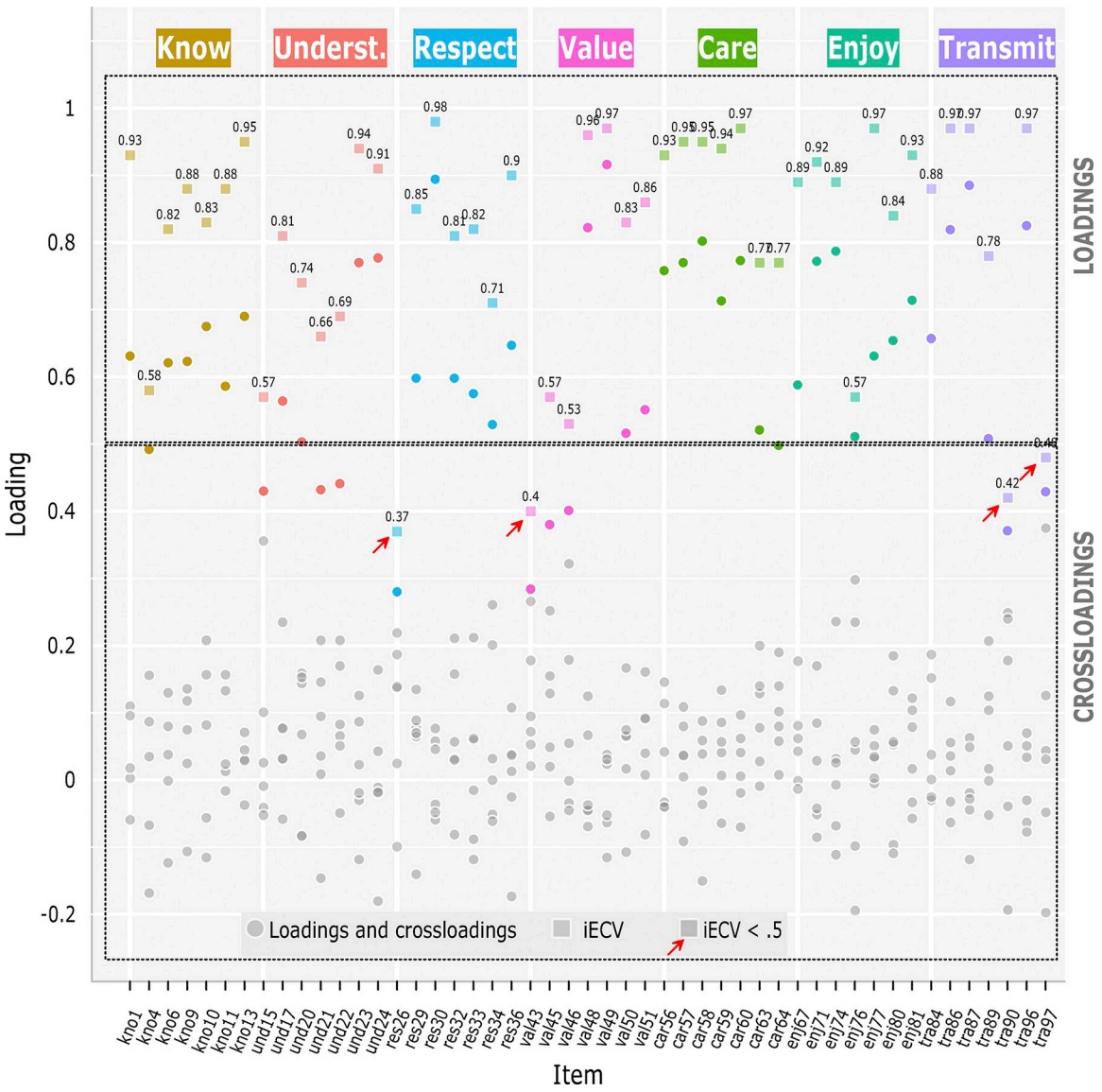

**Fig 5. Loadings and crossloadings.**

**Table 5. Invariance results (gender).**

|  | FP | RMSEA (CI) | CFI | TLI | $\chi^2$ (DF) | AIC | BIC | SRMR |
|---|---|---|---|---|---|---|---|---|
| Configural | 938 | .034 (.032; .036) | .978 | .97 | 3070 (1708) | 140145 | 144536 | .024 |
| Scalar | 553 | .025 (.023; .027) | .985 | .983 | 3028 (2093) | 139929 | 142563 | .028 |

*Note*. FP = free parameters; DF = degrees of freedom; CI = confidence interval

The TICs of the CAR and RES scales showed information profiles that were substantially different from the rest of the scales. The TIC of the CAR scale showed a strong shift to the right of the latent continuum, with maximum information between approximately 0.2 and 2.2 standard deviations above the mean of the latent variable. This implies that the scale discriminates well between people who manifest a medium-high to very high level of CAR, but may have difficulty in accurately detecting individual differences in the low range of the variable. The RES scale, on the contrary, presents a TIC that is strongly shifted to the left of the latent continuum, with maximum information between approximately -2.5 and 0.7 standard deviations around the mean, discriminating accurately between people with medium to low/very low levels on the variable, but with discrimination problems at high and very high levels.

Taking into account the content and purpose of the RES and CAR scales, and the characteristics of the sample, we can conclude that the results described are not unexpected, and do not pose a problem in terms of the validity and usefulness of the measure, for the reasons given below.

The RES scale consists of statements about respect both for heritage as a whole (e.g., "I respect all heritage assets, even if I do not feel identified with some") and for diversity of tastes and opinions (e.g., "I urge others to be respectful of any type of cultural heritage"). Respect for the common good and tolerance of dissent are widespread principles in Western European culture. Thus, it is to be expected that in a questionnaire focused on these values we would obtain a majority of favorable responses and, therefore, maximum discrimination in low areas of the variable (i.e., among people who express neutral or negative attitudes). This expectation is consistent with the results of the analysis, which enables us to conclude that the RES scale:

1. 1. Discriminates well between people who hold attitudes that we might consider normative in Western society (i.e., valuing the common good positively, respecting diversity), and people who deviate from the norm (i.e., valuing neutrally or negatively); and

2. 2. Discriminates well against individual differences in the second group.

The CAR scale, on the other hand, focuses on the evaluation of overt behaviors related to heritage care. It is expected that participants will find the CAR items difficult, and that the discriminative power of the scale will be optimal at medium to high levels of the latent variable, given that:

Unlike the other scales, CAR is organized as a "unipolar dimension" [69], where the negative pole does not represent neglect or mistreatment of heritage, but rather the absence of

**Table 6. Invariance results (age).**

|  | FP | RMSEA (CI) | CFI | TLI | $\chi^2$ (DF) | AIC | BIC | SRMR |
|---|---|---|---|---|---|---|---|---|
| Saturated | 518 | .035 (.034; .037) | .977 | .967 | 2333 (854) | 139656 | 140617 | .018 |
| Invariant | 476 | .036 (.034; .037) | .976 | .967 | 2461 (896) | 139731 | 141962 | .022 |

*Note*. FP = free parameters; DF = degrees of freedom; CI = confidence interval

**Table 7. IRT parameters.**

| Item | α | β₁ | β₂ | β₃ | Item | α | β₁ | β₂ | β₃ |
|---|---|---|---|---|---|---|---|---|---|
| kno1 | 1.995 | -2.596 | 0.928 | 3.691 | car56 | 2.330 | 0.282 | 2.926 | 4.677 |
| kno4 | 1.719 | -3.766 | -0.721 | 2.219 | car57 | 2.621 | 0.100 | 3.050 | 5.421 |
| kno6 | 2.619 | -3.505 | 0.513 | 3.472 | car58 | 2.872 | 1.676 | 4.176 | 6.497 |
| kno9 | 1.957 | -0.966 | 1.586 | 3.595 | car59 | 2.917 | 0.046 | 3.161 | 5.460 |
| kno10 | 2.007 | -1.213 | 1.776 | 4.305 | car60 | 3.144 | 1.026 | 4.193 | 6.603 |
| kno11 | 2.144 | -2.630 | 0.742 | 3.204 | car63 | 2.161 | -0.799 | 1.996 | 4.066 |
| kno13 | 2.916 | -1.894 | 2.388 | 5.593 | car64 | 1.857 | -1.037 | 1.664 | 3.625 |
| und15 | 1.604 | -2.447 | 0.520 | 2.971 | enj67 | 2.334 | -2.977 | 0.299 | 2.780 |
| und17 | 2.564 | -3.790 | 0.557 | 4.057 | enj71 | 2.870 | -3.279 | 0.587 | 3.461 |
| und20 | 1.900 | -1.774 | 1.249 | 4.068 | enj74 | 2.795 | -2.983 | 0.645 | 3.261 |
| und21 | 1.599 | -3.032 | -0.404 | 1.784 | enj76 | 1.741 | -0.211 | 1.935 | 3.666 |
| und22 | 2.197 | -2.796 | 0.855 | 3.870 | enj77 | 2.060 | -1.950 | 0.778 | 2.664 |
| und23 | 1.831 | -2.195 | 0.683 | 3.042 | enj80 | 2.314 | -2.539 | 0.219 | 2.354 |
| und24 | 1.691 | -2.379 | -0.188 | 1.928 | enj81 | 2.917 | -3.309 | 0.711 | 3.492 |
| res26 | 1.236 | -2.668 | 0.169 | 2.626 | tra84 | 2.222 | -1.976 | 0.698 | 2.882 |
| res29 | 1.457 | -2.755 | -0.418 | 1.287 | tra86 | 3.430 | -1.587 | 2.318 | 5.102 |
| res30 | 2.695 | -5.752 | -2.511 | -0.179 | tra87 | 2.498 | -1.253 | 1.737 | 4.012 |
| res32 | 2.265 | -4.480 | -0.422 | 2.650 | tra89 | 2.094 | -1.239 | 1.568 | 3.429 |
| res33 | 1.670 | -4.376 | -1.136 | 1.173 | tra90 | 1.815 | -0.442 | 2.039 | 3.852 |
| res34 | 1.675 | -3.632 | -0.109 | 2.649 | tra96 | 2.206 | -1.363 | 1.287 | 3.254 |
| res36 | 1.692 | -4.542 | -1.726 | -0.108 | tra97 | 1.725 | 0.482 | 2.533 | 4.261 |
| val43 | 1.505 | -1.983 | 0.722 | 2.973 | | | | | |
| val45 | 1.903 | -3.578 | -0.295 | 2.442 | | | | | |
| val46 | 1.868 | -3.803 | -0.562 | 2.246 | | | | | |
| val48 | 2.224 | -2.757 | 0.650 | 3.460 | | | | | |
| val49 | 2.642 | -3.814 | 0.232 | 3.670 | | | | | |
| val50 | 1.758 | -2.745 | 0.369 | 3.181 | | | | | |
| val51 | 2.069 | -2.734 | 0.325 | 2.978 | | | | | |

caring behaviors. Thus, it is logical that the CAR scale should accurately discriminate between people who actively engage in the defense of heritage and those who do not (or do so very infrequently), and should more accurately grade the intensity of active involvement among people in the first group.

The CAR scale, understood as a sample of heritage care behaviors, is limited to actions that take place in social and online media, leaving out of the measurement individual or collective actions that occur exclusively in face to face interactions or by other means. This restriction in the sampling has as an expected consequence a lower observed frequency of caring behaviors, which translates into higher difficulty parameters.

Taking together the results of the CAR scale and the other scales (especially RES), we observe that in this sample the probability of taking actions in favor of heritage is much lower than that of expressing beliefs or "feelings" in favor of heritage. This apparent incongruence was to be expected, given the complex relationship between beliefs and overt behaviors, which should be a logical consequence of the former (see, e.g., Ajzen & Fishbein, 1977 [70]).

## Discussion and conclusions

Q-Herilearn has demonstrated metric guarantees of sufficient validity and reliability as an instrument to accurately measure the processes involved in heritage learning.

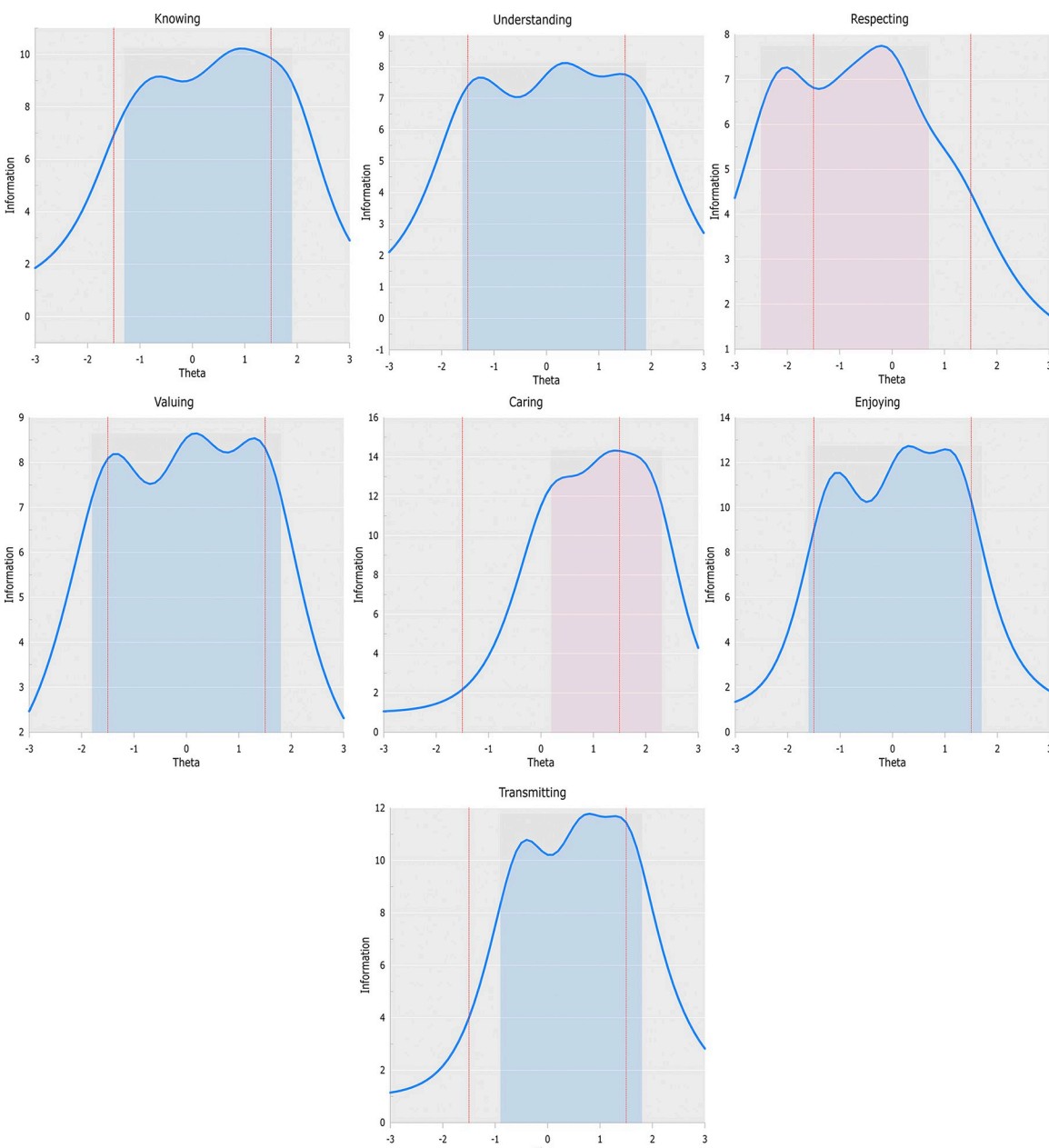

**Fig 6. Test information curves.** A TIC (Knowing). B TIC (Understanding). C TIC (Respecting). D TIC (Valuing). E TIC (Caring). F TIC (Enjoying). G TIC (Transmitting).

Given that there are significant differences in heritage learning outcomes depending on the particular digital medium or mediator in which they have occurred [71], a scale is needed that can be equally used in all digital environments by focusing on structuring dimensions in heritage learning. In this sense, Q-Herilearn would allow comparing the learning outcomes around the same heritage content in different contexts or with different educational mediation strategies.

## Implications

The applicability of the scale encompasses the set of processes and procedures involved in heritage education, i.e., teaching, learning, implementation processes, media/mediators and contexts.

In terms of heritage education—and, in particular, the design of educational program—the 7-dimension structure (which covers the complete sequence of heritage processes) makes it possible to identify the objectives of any heritage education program; each dimension is supported by a verb, and the verbs make up the teaching objectives and, therefore, the heritage learning outcomes. In addition, the items of the scale for each dimension allow to operationally define the learning objectives, so that they can be used individually or in order to relate items from the different dimensions.

In turn, Q-Herilearn will serve as a measurement instrument in the implementation processes of heritage education programs in digital environments, permitting the evaluation of the degree and scope of heritage learning outcomes along the seven dimensions of the HLS, both globally and for each of them individually.

Heritage can be considered as a key element in promoting social cohesion through experiences in virtual environments, in that it equalizes or improves access to opportunities for many people in different geographical areas. In this sense, Q-Herilearn has been calibrated and standardized to be applicable to different contexts, including its translation and adaptation into five other languages (English, French, Basque, Italian and Portuguese).

## Limitations

This study has several limitations. The most important ones refer to the use of a non-probabilistic (incidental) sample. Although the Monte Carlo analysis has shown that the $N$ value used guarantees sufficient precision and statistical power, it should be noted that the non-probabilistic nature of the sample may affect the external validity of the results. In this regard, the three main weaknesses of the study should be noted, which have to do with (a) a limited potential for generalizability, as the sample may not accurately represent the characteristics, diversity or demographics of the population; (b) the selection bias, as the very nature of the data collection instrument (an Internet survey) could result in a portion of the population being overrepresented in the sample; and (c) the lack of variability, as the limited diversity within the sample could restrict the range of responses and reduce the applicability of the results to a broader population. These shortcomings suggest that future research should use a probabilistic sampling methodology based on random selection procedures that provide a higher likelihood of obtaining representative samples from the different populations on which the instrument is applied.

## Future avenues for research

An explanatory model (HPM) has been used to articulate the learning processes in Heritage Education (HLS) that (a) is based on international references, (b) covers a complete cycle in heritage learning and (c) is generalizable and adaptable to different educational designs. The accuracy and consistency of the measure has been demonstrated both in the general scale and in each of the subscales. From here on, the immediate lines of research are geared toward:

Investigating the usefulness of the scale in applied contexts. For example, gauging the extent to which the scale factors are sensitive to change predictably caused by heritage education programs.

Complementing the calibration performed with other analytical approaches (e.g., multi-facet logistic models, network analysis, etc.).

Getting to know which are the most frequent procedures followed by users to learn about heritage in digital environments; i.e., in what ways heritage is learned and what specific learning profiles exist through mixed models (factorial-latent classes).

Applying the full scale in digital heritage learning environments and on different populations to check whether or not there are differences according to socio-demographic traits (e.g., general users, university students, minority groups, people who share different degrees of engagement with heritage, cultural backgrounds, etc.).

Using partial scales—individually or jointly—to measure heritage learning outcomes derived from the implementation of educational designs (these scales would be selected according to the verbs that articulate the objectives of these designs).

Comparing the responses obtained according to the language in which they were answered (i.e., Spanish, English, French, Italian, Portuguese and Basque), or to the bilingual nature of societies with minority languages.

## Supporting information

**S1 Table. Knowing dimension.**
(DOCX)

**S2 Table. Understanding dimension.**
(DOCX)

**S3 Table. Respecting dimension.**
(DOCX)

**S4 Table. Valuing dimension.**
(DOCX)

**S5 Table. Caring dimension.**
(DOCX)

**S6 Table. Enjoying dimension.**
(DOCX)

**S7 Table. Transmitting dimension.**
(DOCX)

**S8 Table. Polychoric correlations.**
(DOCX)

**S9 Table. Monte Carlo analysis results.**
(DOCX)

**S10 Table. Deleted participants and composition of the final sample.**
(DOCX)

**S11 Table. Observed concordance matrix (clarity).**
(DOCX)

**S12 Table. Observed concordance matrix (relevance).**
(DOCX)

**S13 Table. Observed concordance matrix (adequacy).**
(DOCX)

**S14 Table. Observed concordance matrix (dimension).**
(DOCX)

**S15 Table. Average Pairwise proportional agreement for dimension.**
(DOCX)

**S16 Table. Average Pairwise Cohen's kappa for dimension.**
(DOCX)

**S17 Table. Summary of inter-rater agreement analysis for dimension.**
(DOCX)

**S18 Table. Inter-factor correlations.**
(DOCX)

## Author Contributions

**Conceptualization:** Olaia Fontal, Alex Ibañez-Etxeberria.

**Data curation:** Víctor B. Arias, Benito Arias.

**Formal analysis:** Olaia Fontal, Alex Ibañez-Etxeberria, Víctor B. Arias, Benito Arias.

**Funding acquisition:** Olaia Fontal, Alex Ibañez-Etxeberria.

**Methodology:** Víctor B. Arias, Benito Arias.

**Resources:** Olaia Fontal, Alex Ibañez-Etxeberria.

**Software:** Benito Arias.

**Supervision:** Olaia Fontal, Alex Ibañez-Etxeberria, Víctor B. Arias, Benito Arias.

**Validation:** Víctor B. Arias, Benito Arias.

**Visualization:** Benito Arias.

**Writing – original draft:** Olaia Fontal, Alex Ibañez-Etxeberria, Víctor B. Arias, Benito Arias.

**Writing – review & editing:** Olaia Fontal, Alex Ibañez-Etxeberria, Víctor B. Arias, Benito Arias.

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
