## [Decision Letter · Decision Letter 0]

12 Jan 2024

PONE-D-23-37457Q-Herilearn: assessing heritage learning in digital environments. A mixed approach with factor and IRT modelsPLOS ONE

Dear Dr. Fontal,

Thank you for submitting your manuscript to PLOS ONE. After careful consideration, we feel that it has merit but does not fully meet PLOS ONE’s publication criteria as it currently stands. Therefore, we invite you to submit a revised version of the manuscript that addresses the points raised during the review process.

**The manuscript contributes significant advances to contemporary research on approaches and models of assessing heritage learning in digital environments. It is worth highlighting the interesting methodological approach with which the evaluation is approached from a mixed approach with factor and IRT models.**

**Please take into account the minor considerations made by the two reviewers. Your appreciations are highly valuable to enrich some sections of the manuscript in order to increase its quality and expand its potential impact and possible replication in other international contexts.**

**The approach of the proposal is quite original and the methodology is rigorous and quite explicit.**

**A little final effort, it's worth it!**

We look forward to receiving your revised manuscript.

Kind regards,

José Gutiérrez-Pérez

Academic Editor

PLOS ONE

Journal Requirements:

This research has been funded by the Ministry of Science and Innovation, State Research Agency, within the project PID 2019-106539RB-I00, “Learning models in digital environments for heritage education”. Principal Investigators: Olaia Fontal Merillas and Alex Ibáñez Etxeberria.

This research has been funded by the Ministry of Science and Innovation, Next Generation EU (Recovery, Transformation and Resilience Funds), within the project PDC2022-133460-I00, “Heritage education in Spain in the face of the 2030 agenda: heritage literacy plan in digital environments”. Principal Investigators: Olaia Fontal Merillas and Alex Ibáñez Etxeberria.

Please respond by return e-mail so that we can amend your financial disclosure and competing interests on your behalf.

4. We note that you have referenced Petrucco C, Agostini D. which has currently not yet been accepted for publication. Please respond by return e-mail with a copy of your updated manuscript to include to remove this from your References and amend this to state in the body of your manuscript: (Petrucco C, Agostini D. [Unpublished]”) as detailed online in our guide for authors

http://journals.plos.org/plosone/s/submission-guidelines#loc-reference-style.   We can then upload this to your submission on your behalf.

Reviewers' comments:

Reviewer's Responses to Questions

**Comments to the Author**

1. Is the manuscript technically sound, and do the data support the conclusions?

Reviewer #1: Yes

Reviewer #2: Yes

2. Has the statistical analysis been performed appropriately and rigorously? 

Reviewer #1: Yes

Reviewer #2: Yes

3. Have the authors made all data underlying the findings in their manuscript fully available?

Reviewer #1: Yes

Reviewer #2: Yes

4. Is the manuscript presented in an intelligible fashion and written in standard English?

Reviewer #1: Yes

Reviewer #2: Yes

5. Review Comments to the Author

Reviewer #1: First of all, I want to express my appreciation to the authors for carrying out such a comprehensive study, especially methodologically. Actually, we are facing a true investigation that has left nothing to chance in the procedure followed.

However, I would like to make some comments to the authors. These are some comments and evaluations that have arisen after reading the work. This does not imply any criticism, but rather thoughts “out loud” (written), to share.

• In the introduction section, it is indicated that, in the previous studies consulted, questionnaires designed ad hoc have been used mostly, and that these do not present validation processes. This statement is supported by three cited works.

o The first of them, number 8, establishes reliability and validity of the instrument used, so it would not fall within the statement that the authors have made. https://link.springer.com/chapter/10.1007/978-3-319-71940-5_16

o The second, referenced with number 18, effectively presents an innovation and no reference is made to the validation of the two instruments they mention. https://www.emerald.com/insight/content/doi/10.1108/JCHMSD-02-2018-0014/full/html

o Finally, the third, referenced with number 25, presents, like the first study, a reliability analysis. https://ieeexplore.ieee.org/stamp/stamp.jsp?tp=&arnumber=8755497

Therefore, it seems that this statement should conform to proven reality.

• In reference to the sample, I would have avoided including those who appear as residents in Mexico and in “others”, since we do not know if these few data can distort the results. The number of participants without them is so high that it could have been more interesting how the questionnaire is applied in Spain, despite being a non-probabilistic sample.

• Likewise, when talking about the participants' studies, I would have eliminated the six participants with “Primary Education”.

• Surely, despite eliminating the data described above, the results would not have changed, but since there are so few subjects with these variables, it is preferable not to include them, especially having a large sample of participants.

I conclude these words by once again thanking the authors and having been able to read this very interesting work.

Reviewer #2: The present manuscript focuses on the construction, calibration, standardization, and evaluation of the Q-Herilearn instrument for measuring heritage education in digital environments. To achieve this, it employs Item Response Theory (IRT), whose significance in the development and analysis of instrumental quality is sustained throughout the text as a response to the shortcomings of the classical approach (Classical Test Theory, CTT).

The main objective of the study is to address deficiencies in the assessment of processes involved in heritage learning in digital environments by proposing an instrument with robust metric properties. The methodology employed, including exploratory factor analyses and IRT models such as the Graded Response Model (GRM), is appropriate and rigorous for obtaining comprehensive empirical evidence of reliability and validity based on content and internal structure.

Demonstrating the invariance of scores based on participants' gender and age is a relevant aspect that reinforces the instrument's robustness. Additionally, the high item discrimination and test information curves, accurately measuring broad ranges of latent variables, are positive aspects that strengthen instrumental validity.

The proposed applicability for Q-Herilearn is broad at both instrumental and contextual-cultural levels. The 7-dimensional structure (complete sequence of heritage processes) provides a clear and operational approach to heritage learning outcomes. Similarly, the option to use items individually or relate them across different dimensions adds flexibility and versatility to this instrument.

In this regard, as suggested for future research directions, its application is recommended in various formal and non-formal education contexts in virtual environments, based on the verbs defining the didactic objectives (heritage learning outcomes) of each of the 7 dimensions. Since the content of the final items is provided, the application of the complete scale or its subscales to digital heritage learning environments and different populations is facilitated.

Ultimately, the work presents a valuable and significant proposal that will substantially contribute to the literature in the field of heritage education and learning assessment in digital environments.

Finally, as a minor suggestion, and whenever deemed appropriate, a review of the article by Ortega-Sánchez and López-Sanvicente (2023) is recommended, which, we understand, could contribute to the international scientific literature included in this research (https://www.nature.com/articles/s41599-023-01550-z).

6. PLOS authors have the option to publish the peer review history of their article (what does this mean?). If published, this will include your full peer review and any attached files.

Reviewer #1: No

Reviewer #2: **Yes: **Delfín Ortega-Sánchez

---

## [Author Response · Author response to Decision Letter 0]

13 Feb 2024

Dear Dr. Fontal,

* Thank you for submitting your manuscript to PLOS ONE. After careful consideration, we feel that it has merit but does not fully meet PLOS ONE’s publication criteria as it currently stands. Therefore, we invite you to submit a revised version of the manuscript that addresses the points raised during the review process. 

RESPONSE: Thank you for the feedback. We are attaching a revised version in accordance with the suggestions made by the editor and the reviewers.

* The manuscript contributes significant advances to contemporary research on approaches and models of assessing heritage learning in digital environments. It is worth highlighting the interesting methodological approach with which the evaluation is approached from a mixed approach with factor and IRT models. 

RESPONSE: Comments are welcome.

* Please take into account the minor considerations made by the two reviewers. Your appreciations are highly valuable to enrich some sections of the manuscript in order to increase its quality and expand its potential impact and possible replication in other international contexts. 

RESPONSE: We have carefully considered the provided comments and have implemented the necessary modifications to the manuscript.

* The approach of the proposal is quite original and the methodology is rigorous and quite explicit.

A little final effort, it’s worth it! 

RESPONSE: Comments are welcome.

* Please submit your revised manuscript by Feb 26 2024 11:59PM. If you will need more time than this to complete your revisions, please reply to this message or contact the journal office at plosone@plos.org. When you’re ready to submit your revision, log on to https://www.editorialmanager.com/pone/ and select the ‘Submissions Needing Revision’ folder to locate your manuscript file. 

A rebuttal letter that responds to each point raised by the academic editor and reviewer(s). You should upload this letter as a separate file labeled ‘Response to Reviewers’. 

RESPONSE: The rebuttal letter, addressing each of the points raised by the academic editor and reviewers, is appended herewith (this document).

* A marked-up copy of your manuscript that highlights changes made to the original version. You should upload this as a separate file labeled ‘Revised Manuscript with Track Changes’. 

RESPONSE: The document ‘Revised Manuscript with Track Changes’ is attached, in which the changes made are highlighted in red.

*An unmarked version of your revised paper without tracked changes. You should upload this as a separate file labeled ‘Manuscript’. 

RESPONSE: The ‘Manuscript’ document is attached.

*RESPONSE: No changes to the financial disclosure are necessary.

*If applicable, we recommend that you deposit your laboratory protocols in protocols.io to enhance the reproducibility of your results. Protocols.io assigns your protocol its own identifier (DOI) so that it can be cited independently in the future. For instructions see:

https://journals.plos.org/plosone/s/submission-

guidelines#loc-laboratory-protocols.

Additionally, PLOS ONE offers an option for publishing peer-reviewed Lab Protocol articles, which describe protocols hosted on protocols.io. Read more information on sharing protocols at https://plos.org/protocols?utm_medium=

editorial-email&utm_source=authorletters&

utm_campaign=protocols. 

RESPONSE: Not applicable.

*We look forward to receiving your revised manuscript.

Kind regards,

José Gutiérrez-Pérez

Academic Editor

PLOS ONE 

*Journal Requirements: 

Please ensure that your manuscript meets PLOS ONE’s style requirements, including those for file naming. The PLOS ONE style templates can be found at 

https://journals.plos.org/plosone/s/file?id=

wjVg/PLOSOne_formatting_sample_main_

body.pdf and 

https://journals.plos.org/plosone/s/

file?id=ba62/PLOSOne_formatting_sample_title_

authors_affiliations.pdf 

RESPONSE: The manuscript has conformed to the style requirements of PLOS ONE.

*2. Thank you for stating the following financial disclosure: 

This research has been funded by the Ministry of Science and Innovation, State Research Agency, within the project PID 2019-106539RB-I00, “Learning models in digital environments for heritage education”. Principal Investigators: Olaia Fontal Merillas and Alex Ibáñez Etxeberria.

This research has been funded by the Ministry of Science and Innovation, Next Generation EU (Recovery, Transformation and Resilience Funds), within the project PDC2022-133460-I00, “Heritage education in Spain in the face of the 2030 agenda: heritage literacy plan in digital environments”. Principal Investigators: Olaia Fontal Merillas and Alex Ibáñez Etxeberria. 

Please state what role the funders took in the study. If the funders had no role, please state: “The funders had no role in study design, data collection and analysis, decision to publish, or preparation of the manuscript.” 

Please respond by return e-mail so that we can amend your financial disclosure and competing interests on your behalf. 

RESPONSE: The statement “The funders had no role in study design, data collection and analysis, decision to publish, or preparation of the manuscript” has been included in the revised manuscript.

*3. When completing the data availability statement of the submission form, you indicated that you will make your data available on acceptance. We strongly recommend all authors decide on a data sharing plan before acceptance, as the process can be lengthy and hold up publication timelines. Please note that, though access restrictions are acceptable now, your entire data will need to be made freely accessible if your manuscript is accepted for publication. This policy applies to all data except where public deposition would breach compliance with the protocol approved by your research ethics board. If you are unable to adhere to our open data policy, please kindly revise your statement to explain your reasoning and we will seek the editor’s input on an exemption. Please be assured that, once you have provided your new statement, the assessment of your exemption will not hold up the peer review process. 

RESPONSE: The link to the data can be found on lines 585 and 586 of the revised manuscript: https://osf.io/59yr7

*4. We note that you have referenced Petrucco C, Agostini D. which has currently not yet been accepted for publication. Please respond by return e-mail with a copy of your updated manuscript to include to remove this from your References and amend this to state in the body of your manuscript: (Petrucco C, Agostini D. [Unpublished]”) as detailed online in our guide for authors

http://journals.plos.org/plosone/s/submission-

guidelines#loc-reference-style.

We can then upload this to your submission on your behalf. 

RESPONSE: Sorry, there was a mistake in the reference.

The erroneous reference has been replaced by the correct one:

Petrucco C, Agostini D. Walled Cities of Veneto Region: Promoting Cultural Heritage in Education Using Augmented Reality Tools. In: Proceedings of the EDULEARN15. Barcelona, Spain; 2015. https://doi.org/10.13140/RG.2.1.4080.6162.

RESPONSE: The list of references has been reviewed and checked for exact correspondence with the citations in the text. No retracted articles have been included.

*Reviewers’ comments: 

Reviewer’s Responses to Questions

Comments to the Author

1. Is the manuscript technically sound, and do the data support the conclusions?

Reviewer #1: Yes

Reviewer #2: Yes 

2. Has the statistical analysis been performed appropriately and rigorously?

Reviewer #1: Yes

Reviewer #2: Yes 

3. Have the authors made all data underlying the findings in their manuscript fully available?

Reviewer #1: Yes

Reviewer #2: Yes 

4. Is the manuscript presented in an intelligible fashion and written in standard English?

Reviewer #1: Yes

Reviewer #2: Yes 

5. Review Comments to the Author

Reviewer #1: 

First of all, I want to express my appreciation to the authors for carrying out such a comprehensive study, especially methodologically. Actually, we are facing a true investigation that has left nothing to chance in the procedure followed. 

RESPONSE: We appreciate the reviewer’s comments. The clarifications made by him/her will undoubtedly contribute to improving the quality of the article.

*However, I would like to make some comments to the authors. These are some comments and evaluations that have arisen after reading the work. This does not imply any criticism, but rather thoughts “out loud” (written), to share. 

• In the introduction section, it is indicated that, in the previous studies consulted, questionnaires designed ad hoc have been used mostly, and that these do not present validation processes. This statement is supported by three cited works. 

The first of them, number 8, establishes reliability and validity of the instrument used, so it would not fall within the statement that the authors have made. https://link.springer.com/

chapter/10.1007/978-3-319-71940-5_16 

RESPONSE: We concur with the reviewer; study number 8 offers information on validity and reliability, albeit with questionable results attributed to the small sample size (N = 107). Furthermore, certain reliability coefficients (Cronbach’s alpha) have not reached the minimum cut-off point (.701) required for consideration as adequate.

* The second, referenced with number 18, effectively presents an innovation and no reference is made to the validation of the two instruments they mention. https://www.emerald.com/insight/content/

doi/10.1108/JCHMSD-02-2018-0014/full/html 

RESPONSE: Right, the article does not address the processes of validation or calibration for the scores obtained through the mentioned instruments.

* Finally, the third, referenced with number 25, presents, like the first study, a reliability analysis. https://ieeexplore.ieee.org/stamp/

stamp.jsp?tp=&arnumber=8755497 

RESPONSE: Similarly, study number 25 presents a reliability analysis, but the results are also questionable due to the small sample size (N = 62).

*Therefore, it seems that this statement should conform to proven reality. 

RESPONSE: In view of the above considerations, the text has been modified as follows: 

“The studies collected have mostly used ad hoc questionnaires for the specific designs under study in which, in general, the processes of calibration or validation of the scores that were followed are not described in detail. In some cases [8, 25], partial validity and reliability results are provided. In others [18], no information is reported on the metric properties of the instruments. Finally, we have a more complete study [72], in which content validity and inter-observer reliability have been determined. In any case, this ad hoc approach makes it difficult to make reliable comparisons between the results of several studies measuring the same concept.”

*• In reference to the sample, I would have avoided including those who appear as residents in Mexico and in “others”, since we do not know if these few data can distort the results. The number of participants without them is so high that it could have been more interesting how the questionnaire is applied in Spain, despite being a non-probabilistic sample. 

RESPONSE: We concur with the reviewer’s feedback. We have amended Table 1 by removing the variable ‘Residence.’ Nevertheless, it is essential to note that the data should be regarded as purely anecdotal, as no analysis has been conducted with the independent variable being the country of residence.

*• Likewise, when talking about the participants’ studies, I would have eliminated the six participants with “Primary Education”.

• Surely, despite eliminating the data described above, the results would not have changed, but since there are so few subjects with these variables, it is preferable not to include them, especially having a large sample of participants. 

RESPONSE: We agree with the reviewer’s suggestion. We have adjusted Table 1 by excluding the 6 participants with Primary Education.

*I conclude these words by once again thanking the authors and having been able to read this very interesting work. 

RESPONSE: Thank you very much for your comments.

*Reviewer #2: 

The present manuscript focuses on the construction, calibration, standardization, and evaluation of the Q-Herilearn instrument for measuring heritage education in digital environments. To achieve this, it employs Item Response Theory (IRT), whose significance in the development and analysis of instrumental quality is sustained throughout the text as a response to the shortcomings of the classical approach (Classical Test Theory, CTT). 

RESPONSE: We appreciate the reviewer’s thoughtful considerations and are confident that they will significantly contribute to improving the quality of the study.

 

*The main objective of the study is to address deficiencies in the assessment of processes involved in heritage learning in digital environments by proposing an instrument with robust metric properties. The methodology employed, including exploratory factor analyses and IRT models such as the Graded Response Model (GRM), is appropriate and rigorous for obtaining comprehensive empirical evidence of reliability and validity based on content and internal structure. 

RESPONSE: We welcome the reviewer’s comments.

*Demonstrating the invariance of scores based on participants’ gender and age is a relevant aspect that reinforces the instrument’s robustness. Additionally, the high item discrimination and test information curves, accurately measuring broad ranges of latent variables, are positive aspects that strengthen instrumental validity. 

The proposed applicability for Q-Herilearn is broad at both instrumental and contextual-cultural levels. The 7-dimensional structure (complete sequence of heritage processes) provides a clear and operational approach to heritage learning outcomes. Similarly, the option to use items individually or relate them across different dimensions adds flexibility and versatility to this instrument. 

In this regard, as suggested for future research directions, its application is recommended in various formal and non-formal education contexts in virtual environments, based on the verbs defining the didactic objectives (heritage learning outcomes) of each of the 7 dimensions. Since the content of the final items is provided, the application of the complete scale or its subscales to digital heritage learning environments and different populations is facilitated. 

RESPONSE: We find the suggestion of applying the scale in different contexts based on the verbs that define the didactic objectives to be very useful and interesting. In this regard, we will prioritize this future line of research, focusing on working in museums and the school environment, particularly at the primary school stage. This decision aligns with our research focus on teacher training. Additionally, it may be particularly intriguing to explore the integration of the scale with other digital environments created by our team.

*Ultimately, the work presents a valuable and significant proposal that will substantially contribute to the literature in the field of heritage education and learning assessment in digital environments. 

RESPONSE: We appreciate the reviewer’s comment.

*Finally, as a minor suggestion, and whenever deemed appropriate, a review of the article by Ortega-Sánchez and López-Sanvicente (2023) is recommended, which, we understand, could contribute to the international scientific literature included in this research (https://www.nature.com/articles/s41599-023-01550-z). 

RESPONSE: Thank you for the recommendation. We were not aware of the cited article, as it was recently published in 2023. After reading it, we deemed it appropriate to include it precisely as an example of research that conducts instrumental psychometric studies on reliability and content validity. The reference has been added:

72. Ortega-Sánchez D, López-Sanvicente AB. Design, content validity, and inter-observer reliability of the ‘Digitization of Cultural Heritage, Identities, and Education’ (DICHIE) instrument. Humanit Soc Sci Commun. 2023 Feb 13;10(1):1–9. https://doi.org/10.1057/s41599-023-01550-z • Google Scholar

RESPONSE: All figures conform to the format required by PLOS ONE.

---

## [Editor Report · Decision Letter 1]

15 Feb 2024

Q-Herilearn: assessing heritage learning in digital environments. A mixed approach with factor and IRT models

PONE-D-23-37457R1

Dear Dr. Fontal,

We’re pleased to inform you that your manuscript has been judged scientifically suitable for publication and will be formally accepted for publication once it meets all outstanding technical requirements.

Kind regards,

José Gutiérrez-Pérez

Academic Editor

PLOS ONE

---

## [Editor Report · Acceptance letter]

20 Mar 2024

PONE-D-23-37457R1 

PLOS ONE

Dear Dr. Fontal, 

I'm pleased to inform you that your manuscript has been deemed suitable for publication in PLOS ONE. Congratulations! Your manuscript is now being handed over to our production team.

Kind regards, 

on behalf of

Dr. José Gutiérrez-Pérez 

Academic Editor

PLOS ONE